# Operation Analysis of Cultural Heritage Service Ecosystems: Empirical Study Based on Dihua Street and Guansi Shihdianzih Old Street in Taiwan

**Shih-Ling Kuo * and Chun-Liang Chen** 

Graduate School of Creative Industry Design, National Taiwan University of Arts,
New Taipei City 220307, Taiwan; jun@ntua.edu.tw
* Correspondence: kuossuling@gmail.com; Tel.: +886-2-2272-2181

**Abstract:** For urban development worldwide, the revitalisation of cultural heritage and historical buildings is regarded as a strategy for creating jobs, increasing residents' access to local culture, improving their quality of life, and developing the urban economy. The key factor in the revitalisation of cultural heritage and historical buildings is a strategy for developing the urban economy. Through an exploratory study, this paper examined how the cultural service ecosystems of Dihua Street and Guansi Shihdianzih Old Street are created and operated and how actors develop cultural service ecosystems. By presenting a common value proposition, actors can benefit from interactions through an exchange of services, provide cultural services, develop cultural value, and implement full resource integration and value co-creation, thus promoting the cultural brand communication of historical blocks and the sustainability of cultural services. This study further extended the original functions of cultural heritage and analysed the operation of cultural service ecosystems for cultural heritage. The findings of this study revealed that the organisational operational effectiveness of revitalisation and cultural innovation activities in historic districts provided an innovative approach for sustainable development and the economic revival of historical blocks, which can be used as a reference for the sustainability of local culture and economy. In this perspective, this article provides some useful suggestions for stakeholders and policymakers.

**Keywords:** cultural heritage; historical block; resource integration; cultural service; value co-creation



## 1. Introduction

Historical sites and cultural heritage play a critical role in urban renewal. Urban renewal can be regarded as a catalyst for mobilising tangible and intangible heritage. Moreover, it enables cities to develop a cultural heritage-based leisure tourism industry, which creates jobs, provides a source of income, develops the overall urban economy, increases residents' access to local culture, and improves their overall quality of life [1].

Global success cases of urban and rural development demonstrate that the key factor is the ability to develop perfect local industrial ecosystems. This study focused on how actors in the revitalisation and development of cultural heritage should integrate diverse resources across different fields to promote local economic development. The resources of participatory members can be effectively complemented and shared by improving ecosystems, and new business models can be established to benefit all participants. Therefore, this study also focused on how the resources of participatory members can be complemented and shared by improving the ecosystems in historic blocks. Moreover, very few studies have explored cultural heritage revitalisation or cultural services from the perspective of cultural service ecosystems, and international studies lack related empirical examples. Arguments based on traditional commodities are not common internationally, and the problems arising from cultural heritage cannot be covered. Although the concept of 'cultural heritage as a service' has been established internationally, the content and mode of cultural services are

severely misconstrued due to the opposition of different standpoints [2]. Cultural heritage is a multi-dimensional, multi-stakeholder, and multi-disciplinary issue. Thus, it is necessary to adopt an appropriate methodology to analyse the diverse opinions involved [3]. Consequently, cultural heritage service ecosystems can be defined as firms that are held together by various relationships, information, and competencies, including those that can be exploited from other actors in the network of cultural heritage [4–7].

In recent years, cultural and creative industries have rapidly changed, and public sectors have expanded policies related to these industries, such as promoting culture, innovation, and economic vitality [8]. The terms of cultural and creative industries (CCIs) were first mentioned in the policy document called 'Tice six years plan of national construction' published in 1991 [9]. In 1995, the Council of Cultural Affairs (CCA) in Taiwan launched the Total Community Design policy with the slogan 'Culture Industrialization and Industry Culturalization' and the notion of 'Cultural Industry'. It aimed to revitalise the economy in rural areas and towns by using local cultural assets for community formation and social cohesion [10]. In 2010 came the announcement of the 'Law for the Development of the Cultural and Creative Industries' (LDCC), which formalised the definition of the CCIs and also classified the types of industries included [9].

The Central Government of Taiwan (CGT)'s regional and cultural policy implementation was based on the operation of industrial clusters and revitalizing vacant spaces in creative parks. Whereas local cultural support was obtained for creative market promotions and urban marketing methods, the CGT also focused on the importance of developing nationwide regulations and maintaining an organised administration for the operation of cultural and creative industries, and it also implemented a balanced approach to urban and rural development and encouraged the public and private sectors to collaborate in designing and executing industrial policies [8].

The development of policies for CCI clusters in Taiwan is grouped into two policy initiatives: the economic initiative focused on promoting industries such as the digital, information communication and technology, and creative industries to facilitate economic revitalisation of Taiwan and cultural and social initiatives which promote art and cultural assets, traditional techniques, tourism, and post-industrial spatial preservation [9,11–15].

Located in Taipei City, Dihua Street is the centre of Twatutia and has been an important distribution centre of Taiwan for tea and traditional Chinese medicine since the 19th century. In 2010, the Taipei Urban Renewal Office began implementing the Urban Regeneration Station (URS) programme. In light of Landry's concept of the cycle of urban creativity, the Taipei Urban Renewal Office encourages NGOs to settle in URS spaces and use art, culture, and creativity to improve the urban quality of Taipei. Because the best-known cultural and creative organisation in Dihua Street is Sedai Zone Co., many other cultural and creative personnel and organisations have successively settled in Dihua Street, and other local industries have even introduced cultural and creative elements into their business models to transform their business patterns.

The second case is Guansi Shihdianzih Old Street, located in Jhongjheng Road, Guansi Township, Hsinchu County, which was formerly known as Shihdianzih. In 1989, in response to the One Town One Product (OTOP) programme of the Ministry of Economic Affairs, Guansi Town set out to develop the Mesona procumbens Hemsl industry. In recent years, the Council for Hakka Affairs has formulated the 'Implementation Programme for National Hakka Romantic Route along Taiwan Road 3', hoping to stimulate local entrepreneurship and employment. Thus far, several cultural and creative firms have settled in it.

*Research Aims*

As exemplified by the cultural and creative blocks, this study investigated the creation and development of service ecosystems, explored the innovative activities in such blocks, and examined how actors should integrate resources to co-create value. Therefore, this study focused on the following questions:

1.  How are cultural service ecosystems created and operated in historical blocks?

2. How do actors develop cultural service ecosystems through resource integration and value co-creation?

Moreover, regarding the operation of service ecosystems for cultural and creative blocks, this study offers some industrial and policy suggestions and theoretical implications.

## 2. Literature Review

### 2.1. Cultural Heritage

The World Heritage Convention classifies world heritage into cultural and natural heritage sites. Tangible cultural heritage refers to cultural property and buildings with historical, archival, anthropological, archaeological, and artistic values, whereas intangible cultural heritage refers to identifiable representations, knowledge, and skills within specific cultural or social values [16].

Cultural heritage is regarded as a resource for economic development and the creation of activities in cities worldwide [17]. Historical cities have assets possessing both cultural and economic value. The European Commission uses the cultural heritage of cities as key promotional strategies that stimulate short-term and long-term development [18]. This not only extends the lifecycle of heritage but also generates new and different values and supports the development of the local innovation impetus. Cultural heritage enables economic and cycling patterns and can be transformed into new economic, environmental, cultural, and social resources, which are paramount for local and global development.

The preservation and reuse of cultural asset fields are crucial for increasing the reputation and competitiveness of local tourism and the creation of new forms of cultural assets. Historical buildings represent a concrete form of cultural expression that provides an account of past people and things in a particular space. Historical buildings are in line with the goal of economic development if they are integrated with cultural life at the sites of historical buildings. In addition to the cultural connotation and value of the buildings themselves, it is important to reuse historical buildings by taking appropriate measures such as combining traditional historical and cultural memories with the industrial design strategy of 'cultural creativity' and 'innovative experience', identifying the existing cultural elements, proposing relevant plans, creating multi-functional spaces, exerting the spatial, cultural, and thematic advantages, and increasing the depth of content to attract consumers through the design of situational experiences.

### 2.2. Local Culture Development

Culture can be used as a force to maintain group relations and is an essential element of a community network [19]. Hence, culture can be defined as a system of meanings including symbols, rituals, values, and ideologies. For residents to turn culture into a fundamental resource of knowledge and dialogue for the formation and maintenance of communities, it is imperative that communities consistently share sources, practice and develop community culture, and maintain a sense of community identity and resources for collective action. Therefore, culture serves to develop a sense of local identity, promotes local solidarity, and influences the confidence of residents in solving problems jointly. Furthermore, the belief in a common ideal arises from the interaction of different opinions [20].

In addition, culture is an important asset that represents urban images and memories and symbolises a sense of local belonging [21]. The sense of identity provided by the local culture can assist individuals in playing different roles. Mobilisation further enables individuals to develop a strong sense of identity with cultural connotations, thus allowing cultural resources to build the sense of cultural identity required for collective action in a community [22].

Examining the current phenomenon of cultural development in Taiwan from the perspective of institutions of cultural and spatial governance reveals that cultural policies play a positive role in fostering local cultural consciousness, leading to cooperation between the public and private sectors with an emphasis on the assistance of governmental plans, the participation of private firms, and the operation of local community organisations,

shaping of local cultural images, and regeneration and cohesion of cultural consciousness. To enroot the local cultural industry, it is imperative that people enjoy the culture and feel its distinct, unique characteristics. Building a close link between daily life and the cultural industry is a matter of necessity and the ultimate goal of a cultural industry.

### 2.3. Service-Dominant Logic, Cultural Value, and Cultural Services

The revitalisation of cultural heritage and historic areas is in full swing under the guidance of cultural policies in many countries. Thus, numerous appropriate cultural services have emerged. UNESCO defines cultural services as activities aimed at satisfying cultural interests or needs that do not represent material objects per se but usually include supporting measures (e.g., actual cultural behaviours and activities) provided by governments and non-governmental organisations (NGOs, quasi-public institutions, and firms). In the process of sociocultural innovation, the participation of actors enhances the value of innovation and extracts value from a city's resources of material, immaterial, and intangible assets [23]. In turn, this process stimulates a creative atmosphere that enables innovation.

According to service-dominant logic, services are the basis of exchange and represent an entity that applies its knowledge and expertise (e.g., expertise or operational resources) to benefit another entity [6]. The interaction between two entities leads to the co-creation of services that are continuously exchanged to generate value primarily through the active participation of actors and the provision of operational resources for exchange [6]. Cultural value is generated through the exchange of different services or from experiential interactions between actors [24], thus leading to the process of value creation. Accordingly, the connotation of cultural goods shifts from intrinsic value to use value, while value arises from the co-production of actors and active transformation [25,26]. Many studies have argued that actors drive and participate in the process of creating cultural value [2,27]. In this regard, cultural value is created not only by experiencing pleasant additional services but also by actors' participation in the creation of value. Accordingly, cultural value is not predetermined by goods but rather created through the interaction of goods and actors. Customer experience depends not only on the intrinsic value of cultural goods but also on the service experience shaped by service providers.

The interactions between different actors in cultural services can add to their value and consequently expand the boundaries of the concept of 'value'. In particular, service networks can provide more diverse aspects and an actor perspective for cultural goods to boost cultural value. Actors who co-create value in a constellation network share resources through cultural services. The process of value co-creation by actors of cultural services can nourish and multiply cultural values because all actors are committed to sharing their resources. A cultural service system integrates people, technologies, processes, and information [28]. Furthermore, it is self-adaptive through mutually beneficial interactions and resource integration [29], thus promoting resource integration [29–31]. In addition to shaping a constellation network for actors, a cultural service system interacts through a platform to co-create value and integrate actors' information and experience [30]. Cultural service ecosystems are formed when the resources of actors are integrated and interconnected through common institutional arrangements and when actors co-create value while sharing the experience of services. A perfect institutional framework for the development of the local cultural industry is built through the sustainable operation of culture and a harmonious network and alliance relationship between diverse actors. In this sense, historical blocks not only shape an interactive environment but also offer a service environment that provides experience, allows experience sharing, re-creates cultural value, and even enhances their cultural value.

Traditionally, cultural heritage is distinguished into tangible cultural heritage and intangible cultural heritage. Tangible cultural heritage includes culturally significant buildings, structures, sites, and places, whereas intangible cultural heritage includes ideas, practices, beliefs, traditions, and values that serve to unite a particular group of people [31]. The above 'intangible' perspective of cultural heritage does not fully convey any conceptual

change. Cultural value itself does not exist in cultural goods or services but must be extracted from potential cultural connotations to provide the common value proposition shared by actors, meet their expectations of value co-creation, and strengthen the value-adding process in which potential actors actively participate during value creation. Thus, cultural value is not predefined but emerges from the interactions between actors. However, traditional marketing methods for products and services may not present cultural value in its entirety and may even be prone to banalising cultural value [2]. Accordingly, it is worthwhile to examine how the actors of historical blocks can interactively extract and increase the cultural value of historical blocks, subsequently developing cultural service ecosystems for historical blocks through resource integration and value co-creation.

## 3. Materials and Methods

### 3.1. Methodology

Few studies have explored how tangible and intangible heritage are integrated as a strategy for creating jobs, developing the overall urban economy, increasing residents' access to local culture, and improving their overall quality of life. Thus, this study adopted the exploratory study approach [32,33], as it allowed us to explore the complexity of innovation cases and identify emerging concepts [32–34]. The case study method is suitable for issues that are rarely discussed, and by observing the actual environment, this method examines the exploratory issues (e.g., why and how) and analyses the process and cause of a specific case, thus providing a basis for other studies [33]. Therefore, the case study method was adopted in this study with a focus on one or several instances of certain social phenomena and obtaining large quantities of detailed data from a few cases [35]. The acquired content and data could be analysed and consolidated to determine the origin, cause, background, impact, and significance of related events. The selection of study cases followed the logic of replication, analysing the observed phenomena, and developing an inductive method for interpretation [33]. This study explored the following research question: 'How can the actors of Dihua Street and Guansi Shihdianzih Old Street use cultural heritage effectively to establish cultural value propositions, provide cultural services through service exchange, promote brand communication, achieve value co-creation through resource integration, and create cultural ecosystems?' Long-term in-depth interviews were conducted to obtain rich and detailed data on the respondents' backgrounds, actors, and the relationship between historical events and their decisions, thus facilitating the preliminary exploration of related theories [36].

Following Langrish [37], the selection of cases in this study was based on three criteria, namely 'representative', 'the one next door', and 'the best practice'. In this study, Dihua Street was awarded the third prize in the most competitive 'Charming Characteristic Group' for the 'Distinctive Commercial District Award 2020' held by the Small and Medium Enterprise Administration of the Ministry of Economic Affairs. According to a survey conducted by the Department of Information and Tourism of Taipei City, Twatutia was the third most popular attraction in Taipei for international tourists, surpassing the popularity of the Beitou Spring Resort for the first time. The Guansi Shihdianzih Old Street is now the most popular scenic spot for international tourists in Hsinchu County. Because both Dihua Street and Guansi Shihdianzih Old Street met the above selection criteria, they were selected as the cases for this study.

This study integrated the exposition of the service-dominant logic by Vargo and Lusch, an analysis framework for a service system by Grobbelaar et al. [38], and a value co-creation system for a cultural ecosystem constructed by Ciasullo et al. [39]. Accordingly, this study examined how the actors in the two cases exchange services by establishing value propositions, surveying the market structures, and integrating cultural resources to provide cultural services, thus communicating cultural brands, integrating resources in the process of value co-creation, and developing cultural service ecosystems. Figure 1 shows the structure of the initial analysis.

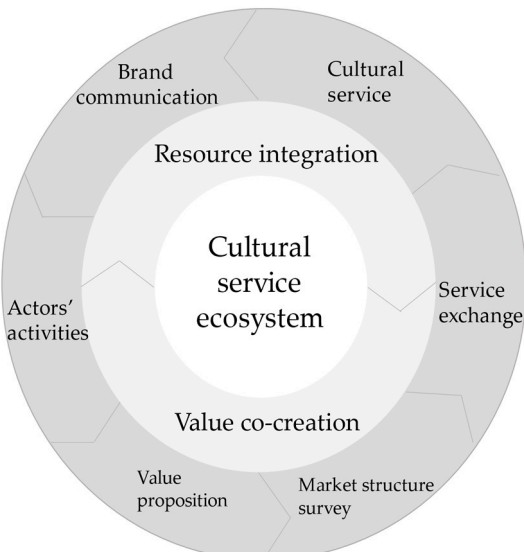

**Figure 1.** Cultural service ecosystem in the context of value co-creation (Adapted with permission from Ref. [38]. Copyright 2016 Taylor & Francis Online and [39]. Copyright 2018 Scientific Research Publishing).

*3.2. Data Acquisition*

The main data were sourced from the academic literature and interview records. To conduct this study, both internal and external personnel were surveyed to collect diverse opinions. The academic literature is based on primary and secondary data, and the collected content and data were analysed and consolidated to determine the origin, cause, background, impact, and significance of related events. There are six possible sources of evidence for case studies: documents, archival records, interviews, direct observation, participant observation, and actual artefacts [33]. Secondary data collected for this study included public authorities' administrative plans, administrative reports, working reports, annual reports, news releases, publications, official websites, and other documents (e.g., newspapers and Internet news media, published monographs, journal articles, and dissertations). Personal interviews were the main source of the primary data. A total of 56 respondents were interviewed for this study, including 2 experts with an average of 25 years of professional experience, 4 members from local development associations, 12 local cultural and creative personnel, 6 consumers, and 4 local residents. The respondents were coded as follows: experts (E1–E2), association members (A1–A4), cultural and creative personnel (I1–I12), consumers (C1–C6), and local residents (R1–R4). The semi-structured interviews focused on themes such as company operation, marketing of innovative services, consumption patterns, changes in the industry structure, policy implementation, and future development. Finally, personal interviews were temporarily ended based on the principle of theoretical saturation [40], with a cumulative verbatim transcription of approximately 300,000 Chinese characters. The interview period for this study was from 6 August 2020 to 3 July 2021, with 49 subsequent interviews conducted to collect additional data and address nuances. The collected data were manually analysed without using any software. To survey the operation of all firms and cultural and creative personnel, direct observations were conducted on their experience services and product sales. The interview questions are briefly listed in Table 1.

*3.3. Data Analysis*

Yin [33] proposed a three-step approach to coding and data analysis: summarise the main findings, analyse the unique characteristics of each case, and conduct cross-case analysis. Through cross-case analyses, researchers can obtain findings from each case and compare these with the findings of other cases to identify new patterns and themes [34]. As similarities and differences between cases were identified in this study, they were further

analysed through repeated interviews. Each service system was analysed according to the following main structure: actors' activities, presenting value propositions, surveying the market structures, exchanging services, establishing cultural value, providing cultural services, brand communication, resource integration, and value co-creation. Diagrams were used to draw the structural hierarchy and to compare several possible structures [33]. Based on the emerging themes and structures, we developed a preliminary view of the relationship between different dimensions [32]. Case studies were conducted using a variety of data collection methods (e.g., interviews, literature data, and direct observations) [41]. Multiple sources of data or triangulation points can be used to ensure internal validity and to minimise potential bias in drawing conclusions [33]. Two members of the research team participated in each interview and took notes to improve the content and validity of the interviews [32]. Table 2 lists the categories of the respondents surveyed in this study.

**Table 1.** Interview questions.

| Attributes | Questions |
|---|---|
| Organisational aspect | 1. Is the concept of product design or service design based on the local culture of cultural and creative blocks? <br> 2. How does your company market your product or service? |
| Economic aspect | 1. Did you evaluate whether the values were similar to those of cultural and creative firms? <br> 2. Did you consider cooperating with other companies in the neighborhood at that time? <br> 3. What is the image or opinion that your company hopes to convey to consumers? <br> 4. Based on your observation and understanding, after cultural and creative firms entered the cultural and creative blocks, do consumers identify the local culture within the cultural and creative blocks? <br> 5. Are there any related cultural activities to promote consumers' understanding of the local culture? |
| Sociocultural aspect | 1. After the establishment of cultural and creative firms, what has changed in the cultural and creative blocks? <br> 2. Do you agree with the cultural and creative firms' business model? |

**Table 2.** Profile of respondents and dates of interviews.

| Code | Category | Internal or External Resource | Firm | Title | Data Source | Date of Interview |
|---|---|---|---|---|---|---|
| I1 | Industry expert | Internal | Generation Block Inc. | CEO Secretary | Interview | 6 August 2020 |
| I2 | Industry expert | Internal | All Black | Leader | Interview | 4 November 2020 |
| I3 | Industry expert | Internal | Peacock Bistro | Leader | Interview | 6 November 2020 |
| I4 | Industry expert | Internal | Primrose | Leader | Interview | 30 March 2021 |
| I5 | Industry expert | Internal | DdC | Leader | Interview | 23 February 2021 |
| I6 | Industry expert | Internal | Luguo Café | Leader | Interview | 9 April 2021 |

**Table 2.** *Cont.*

| Code | Category | Internal or External Resource | Firm | Title | Data Source | Date of Interview |
|------|----------|-------------------------------|------|-------|-------------|-------------------|
| I7 | Industry expert | Internal | Li Ting Xiang | Deputy general manager | Interview | 16 April 2021 |
| I8 | Industry expert | Internal | Shihdianzih 69 Organic Bookstore | Owner | Interview | 3 February 2021 |
| I9 | Industry expert | Internal | Lo Ink House | Leader | Interview | 2 February 2021 |
| I10 | Industry expert | Internal | Experimental Life Shop 1 of Guansi Relationship | Leader | Interview | 5 February 2021 |
| I11 | Industry expert | Internal | Dazihzai | Leader | Interview | 9 February 2021 |
| I12 | Industry expert | Internal | DT52 | Leader | Interview | 18 February 2021 |
| A1 | Industry association | External | | Former president | Interview | 3 February 2021 |
| A2 | Industry association | External | | Former director-general | Interview | 2 February 2021 |
| A3 | Industry association | External | | President | Interview | 9 February 2021 |
| A4 | Industry association | External | | Supervisor | Interview | 3 July 2021 |
| E1 | Expert | External | | | Call interview | 16 June 2021 |
| E2 | Expert | External | | | Call interview | 6 June 2021 |
| C1 | Consumer | External | | | Interview | 15 December 2020 |
| C2 | Consumer | External | | | Interview | 5 February 2021 |
| C3 | Consumer | External | | | Interview | 17 February 2021 |
| C4 | Consumer | External | | | Interview | 18 February 2021 |
| C5 | Consumer | External | | | Interview | 19 February 2021 |
| C6 | Consumer | External | | | Interview | 21 March 2021 |
| R1 | Resident | Internal | | | Interview | 16 February 2021 |
| R2 | Resident | Internal | | | Interview | 18 February 2021 |
| R3 | Resident | Internal | | | Interview | 18 March 2021 |
| R4 | Resident | Internal | | | Interview | 12 May 2021 |

## 4. Case Profile

### 4.1. Case Description

Located in Taipei City, Dihua Street is the centre of Twatutia and has been an important distribution centre of Taiwan for tea and traditional Chinese medicine since the 19th century. In 2010, the Taipei Urban Renewal Office began implementing the URS programme. In light of Landry's concept of the cycle of urban creativity, the Taipei Urban Renewal Office encourages NGOs to settle in URS spaces and use art, culture, and creativity to improve the urban quality of Taipei. Because the best-known cultural and creative organisation in Dihua Street is Sedai Zone Co., many other cultural and creative personnel and organisations have successively settled in Dihua Street, and other local industries have even introduced cultural and creative elements into their business models to transform their business patterns.

The second case is Guansi Shihdianzih Old Street, located in Jhongjheng Road, Guansi Township, Hsinchu County, which was formerly known as Shihdianzih. In 1989, in response to the One Town One Product (OTOP) programme of the Ministry of Economic Affairs, Guansi Town set out to develop the Mesona procumbens Hemsl industry. In recent years, the Council for Hakka Affairs has formulated the 'Implementation Programme for National Hakka Romantic Route along Taiwan Road 3', hoping to stimulate local entrepreneurship and employment. Thus far, several cultural and creative firms have settled in it.

*4.2. Cultural and Creative Block in Dihua Street*

In 2010, Sedai Zone Co. settled in Dihua Street and founded Art Yard in the local landmark shop, Watsons Pharmacy, followed by the founding of a number of artistic yards to mentor cultural and creative start-ups. To date, more than 40 brands or cultural and creative organisations have settled in Dihua Street. There are three modes of cooperation with the cultural and creative organisations: assisting them in house space planning, providing the point of sale system, providing human resources, legal, and financial consulting services, developing products cooperatively and generating revenue together, and developing a closer partnership (e.g., cooperative investment or joint venture). Because of the clustering effect of Art Yard, many other cultural and creative personnel and organisations have successively settled in Dihua Street, and other local industries have even introduced cultural and creative elements into their business models to transform their business patterns and attract more young customers.

In 2000, the Taipei City government designated the surroundings of Twatutia as the 'Twatutia Historic Special Zone' (Taipei's first historical special zone) and promulgated the master plan and other detailed plans. In addition to assisting the reconstruction and renovation of old houses and developing public facilities, the Twatutia Storytelling Workshop (Visitor's Centre) was set up to provide tour guides and consultancy services and organise free experiential activities. Taipei Mayor Wen-Je Ko has plans to turn Twatutia into the fifth wall-less city museum in Taipei City. Furthermore, the Department of Commerce of Taipei is implementing the Traditional Shop Rejuvenation Project to assist old shops in highlighting their unique characteristics and assisting with rejuvenating old shops. In addition to the reconstruction and renovation of old houses in Dihua Street, the key to the revitalisation of cultural heritage lies in leveraging local cultural capital, integrating community resources, and supporting the transformation of local traditional industries. In addition, the Small and Medium Enterprise Administration of the Ministry of Economic Affairs implemented the Small Business for Township Revitalisation (SBTR) programme to mentor urban and rural revitalisation. The SBTR programme is intended to embed industries locally, promote the return of previous residents, and stimulate their career growth in their hometowns. The SBTR programme is implemented to help local old firms transform their businesses, and experiential activities are held to attract young customers and encourage young people to inherit old firms.

Local shops with different types of businesses often collaborate to organise experiential activities. In addition, local tourism practitioners organised in-depth tours to learn about the local history and culture.

*4.3. Guansi Shihdianzih Old Street*

Owing to the support of the Ministry of Culture, the continuous promotion of Hsinchu County, and active participation of Guansi's local associations, a number of cultural and creative firms have settled in Guansi Shihdianzih Old Street, thus preliminarily shaping some cultural and creative industry clusters. Afterwards, the Cultural Affairs Bureau of Hsinchu County commissioned Fantasy Story to transform Guansi Shihdianzih Old Street and carry out basic repairs on its old houses. In the first two years, Fantasy Story was commissioned by property owners to rent out old houses to cultural and creative firms. After this contract expired, the cultural and creative firms could rent old houses directly

from the property owners. Initially, 10 cultural and creative organisations settled in the old street.

Some communication associations have been founded, including Guansi Township's Native Culture Association of Hsinchu County, which is engaged in local cultural and historical surveys, and the Guansi Art Town Development Association of Hsinchu County. In addition, the association provides tourist guide services and develops tour routes to promote local culture. Based on Jhongjheng Road, where Guansi Shihdianzih Old Street is located, the Guansi Art Town Development Association is committed to revitalising and regenerating traditional buildings and encourages the participation of local residents to boost development and achieve the industrialisation of local culture.

## 5. Results and Discussion

Based on the initial analysis framework mentioned above, this study examined how the actors in the two cases exchanged services by establishing value propositions, surveying the market structures, and integrating cultural resources to provide cultural services, thus communicating cultural brands, integrating resources in the process of value co-creation, and developing cultural service ecosystems. In addition, this study further analysed how the cultural service ecosystems in the two cases were created and operated.

### 5.1. Creation of Cultural Service Ecosystems in Historical Blocks

Actors' Activities and the Survey of the Market Structure and Value Propositions. In cultural innovation activities, actors seek value from tangible and intangible cultural heritage through a survey of market structures and exchange of services to increase the value of cultural innovation, thus mutually benefiting themselves and the cultural heritage they represent.

The commercial district of Dihua Street is located in a declining older urban area, and local property owners are not willing to rent out their houses to non-local people. Therefore, very few cultural and creative organisations have settled in Dihua Street. In addition, most consumers in Dihua Street are elderly individuals who want to buy traditional Chinese medicine and native products, whereas few are young people or overseas tourists. After settling in Dihua, Sedai Zone Co. started to rent street houses from local property owners, introduced start-up cultural and creative firms, and rented out houses on the street to cultural and creative firms in a shop-in-shop manner. In addition, it continues to provide indispensable assistance for start-up firms, thus lowering the access threshold of Dihua Street and resulting in a clustering effect of cultural and creative firms that has altered the original commercial structure of the commercial district. Consequently, Dihua Street attracts a large number of young consumers and overseas tourists every year.

### 5.2. Service Exchange

In addition to operating its own well-known design brand, Sedai Zone Co. actively introduces the design works of new generations, hoping to attract consumers' attention to new-generation designers' works, increase their exposure and visibility in the market, and promote the clustering effect of culture. While supporting the steady growth of new brands and developing cultural clusters, the company also contributes its own knowledge and expertise to the exchange of services.

The cultural and creative firms in Guansi Shihdianzih Old Street have adopted different business models. Initially, the Cultural Affairs Bureau of Hsinchu County commissioned Fantasy Story to renovate old street houses and attract investment. At the end of the subsidy programme, it withdrew from the old street, and the Guansi Art Town Development Association helped new cultural and creative firms apply for subsidies from public authorities and convened them for various activities instead. With the initial guidance and assistance of the association, local cultural and creative organisations have gradually grown from start-ups to stable and mature operations. Local firms desired to revitalise the local economy but lacked consensus over long-term goals. In addition, each member of the association had to

undertake several jobs concurrently and thus failed to fully devote themselves to the duties of the association. Hence, the association's function of brainstorming gradually diminished, making it no longer important to the old street. Local firms, mainly in collaboration with other shops or organisations, hold various events.

At present, Shihdianzih 69 Organic Bookstore has transformed its business from a bookstore and B&B services into community construction and cultural services on a resource-sharing basis. The bookstore has recruited a team of mothers who volunteer as storytellers and offers training courses to enhance their storytelling skills. Compared with metropolitan areas, Guansi Township lacks sufficient social and cultural resources and encounters various problems associated with family dysfunction. The bookstore offers an effective solution for certain family dysfunction problems. In the second year after its founding, it launched the 'Shared Good Canteen', which holds a weekly meal for residents, creating a space for them to discuss the development vision of the community and exchange ideas, as well as providing a place for solitary elderly persons to interact with others.

According to the service-dominant logic, services are the basis of exchange and function as entities that apply their knowledge and skills to benefit another entity [26]. The business model of Generation Block Inc. or the philosophy of resource sharing of Shihdianzih 69 Organic Bookstore is in line with the view of Lusch and Vargo [42], in which services refer to the use of resources to create benefits for other actors and are the basis of all economic exchange activities. With a core role, services are defined as the application of special skills (active resources) to create benefits for other individuals or the individuals themselves through actions, processes, and performance [26]. Accordingly, this study proposes the following proposition:

**Proposition 1.** *Through active participation, actors in historical blocks exchange operational resources, co-create services, and exchange services continuously, thus establishing value propositions.*

### 5.3. Operation of Cultural Service Ecosystems in Historical Blocks

5.3.1. Establishment of Cultural Value and Cultural Services

In recent years, culture has become a major factor in urban competitiveness and has been emphasised to achieve cities' social, economic, and political goals [21,43]. In the urban production system, the combination of capital and the sense of identity has a positive economic effect, and cultural activities and industries are an important part of the urban development strategy for urban renewal and improvement of tourism benefits [21]. The role of culture has shifted from a constituent part of social and political heritage to an effective tool for promoting urban brands and maintaining social cohesion [44].

Cultural service systems not only shape a star network of actors but also interact through platforms to co-create value and integrate their information and experience, thus integrating diverse resources (e.g., people, technologies, processes, and information) through mutually beneficial interactions [26,30,45].

Based on the existing cultural elements of Dihua Street, Sedai Zone Co. has created a multi-functional space to revitalise the historical block, integrated cultural services through community influence, and helped start-up firms find ways to thrive. This service platform plays a bridging role between the brand and the market, helping brands increase their market presence. The connotation of cultural products is refocusing from intrinsic value to use value, where value is an outcome of co-production and active transformation of actors [6,25]. By offering a small tour of Hakka life and culture, Guansi Shihdianzih 69 Organic Bookstore invites local farmers to teach traditional farming techniques, allows tourists to experience traditional buffalo farming methods, and promotes an organic rice brand entitled 'Buffalo-farmed Rice'. As a result, the local farming concept and patterns have changed, and consumers have become more habituated to eating organic rice.

### 5.3.2. Brand Communication

Every year, the Taipei City government allocates considerable funds to the promotion of Dihua Street. For example, the Twatutia fireworks display and the shopping marketplace for the spring festival attract hundreds of thousands of tourists annually. The Xia-Hai City God Temple attracts nearly 50,000 overseas tourists annually from Japan, Hong Kong, southeast Asian countries, Europe, and the United States. Furthermore, the temple employs foreign tour guides to present the long history and religious culture of Dihua Street to overseas tourists from different cultural backgrounds. JTB Corporation, Japan's largest travel agency, also regards the Xia-Hai City God Temple as a must-see scenic spot on its tour itinerary. The 'Island Walk' tour is often offered to the foreign guests of Taiwan's public authorities. In the annual Twatutia International Art Festival held by Sedai Zone Co., the cross-dressing parade with the historical background of the 1920s offering colourful activities is organised, focusing on the theme of the culture and history of Dihua Street. In addition, foreign artists are invited to attend the local art festival, thus promoting cultural exchange and presenting the history and culture of Dihua Street. Under the guidance of the Department of Commerce of Taipei and the sponsorship of the Small and Medium Enterprise Administration of the Ministry of Economic Affairs, the Dihua Commercial District Promotion Association holds the annual 'Herbal Medicine Party', which aims to help young people understand the miracle of traditional Chinese medicine and native products using different sense organs. Through international marketing, Dihua Street has become a stage for cultural branding. Today, 'Dihua Street' is established as an internationally famous brand.

Here, Lo Ink House is taken as an example. It offers B&B services, allowing tourists to appreciate the architectural beauty of the century-old, triangle-shaped courtyard. It also arranges guided tours of famous scenic reports of Guansi Town, holds DIY activities, and develops in-depth tour routes that seek to slowly appreciate the local culture. Moreover, local erhu musicians were invited to write a musical drama based on the life experiences of the ancestors of the Lo family, which was performed at the National Concert Hall and Lo Ink House. The amateur theatrical troupe formed by local residents also presents Hakka culture through theatrical performances.

Historical sites and deserted buildings carry the memories of local history, culture, and residents and are essential local assets. Hence, it is imperative to preserve the historical remains of old space, excavate and highlight local characteristic resources, shape charming towns, increase the competitiveness of old towns, link historical and cultural tourism with economic development, create a clustering effect of the various concerned parties, and create positive symbiosis and sharing with the surroundings. The revitalisation of fields not in use not only increases the flexibility in the use of idle space and facilities but also emphasises the importance of being enrooted in local areas, having close connections with residents, and enhancing the shared sense of identity. A typical case analysis of experimental fields can help historical blocks shape local cultural brands and replicate relevant experiences suited to their local characteristics, thus facilitating their development. Accordingly, this study presents the following proposition:

**Proposition 2.** *To revitalise cultural heritage, such as historical blocks, it is imperative that the actors provide cultural services, create cultural brands, implement brand communication, and promote local economic development.*

### 5.4. Resource Integration and Value Co-Creation

### 5.4.1. Value Co-Creation by Actors in Historical Blocks

Value is co-created by firms, employees, shareholders, customers, government agencies, and other entities associated with any given transaction [4]. Value is created when service users (i.e., consumers) interactively exchange resources, services, and information

with service providers (i.e., producers) and further integrate their knowledge and expertise into the experience of use [46–48].

To create a community of value, Sedai Zone Co. recruited appropriate teams for five Twatutia industries (tea, clothes, agricultural products, Chinese opera, and architecture) and stressed the importance of creating a good ecosystem, namely by developing good small clusters in Dihua Street and then merging them into large clusters. Traditional street houses with rich historical and cultural connotations are among the most important assets of Dihua Street. Once Dihua Street prospers, the street house cooperation model proposed by Sedai Zone Co. will benefit the property owners, who in turn will be motivated to rent out more street houses for places of business. Alternatively, Sedai Zone Co. can negotiate rentals with property owners on behalf of cultural and creative firms to prevent a continuous rise in housing rental costs. As a result, start-up cultural and creative firms can be free from the pressure of rents being raised and be more devoted to their primary business. The ultimate goal is to attract young cultural and creative personnel, produce a clustering effect, and achieve the sustainable development of Dihua Street.

5.4.2. Resource Integration by Actors in Historical Blocks

To revitalise Dihua Street, the Taipei City government has allocated considerable funds and resources. However, many local residents experience difficulty adapting to the ongoing tourism-oriented transformation in recent years. Not all residents agreed concerning the alleged success of the settlement of Sedai Zone Co. in Dihua Street. Because the turnover of counters is very fast, the landlords can make money merely by renting their houses out, but the cultural and creative activities are the most painstaking.

Although annual large events in Dihua Street are fully funded by the Taipei City government, they are not commercially integrated with the cultural and creative firms in Dihua Street. Hence, these events attract a flood of people but do not produce economic benefits as expected, even causing an extra burden to local residents. The settlement of cultural and creative firms has changed the industrial structure of Dihua Street and has caused a rise in the cost of shop rentals. As a result, the original local industries failed to survive and had to withdraw from Dihua Street.

In contrast, public authorities entrusted Fantasy Story with the operation of Guansi Shihdianzih Old Street, underpinned by a 2-year government subsidy programme. When the programme term expired, Fantasy Story withdrew from the old street, and instead, the Guansi Art Town Development Association assisted start-up firms to operate or apply for subsidies from public authorities. Because local residents failed to reach a consensus on the development vision of the old street, and subsidies and resources were not evenly distributed, both the cohesiveness and cooperative willingness between the cultural and creative firms were very low. As a result, the Guansi Art Town Development Association gradually failed to play its role. In addition, local property owners were not willing to rent out their houses. It is somewhat difficult to form cultural and creative clusters if business places are in short supply over a long period.

Moreover, local public authorities attach importance to large-scale public construction but lack funding for the revitalisation and revival of the old street (e.g., maintenance of old houses, hardware facilities, and promotion and marketing). One solution would be to have property owners or local cultural and creative firms maintain old houses on their own or organise events to publicise them. Following the example of other counties or cities, the old street introduced Fantasy Story to assist with operations. However, as Fantasy Story was not a locally rooted firm, it could not continue to operate in the old street once the 2-year subsidy programme expired. The resources public authorities allocated to the cultural development, promotion, and publicity of the old street were very limited. Notably, the problem was not the spending power of tourists but rather the fact that an insufficient number of tourists came that was needed to generate enough revenue for local shops to survive. Consequently, local cultural and creative personnel had to take up other jobs concurrently to keep their main businesses from going under. Due to these reasons, along

with the rise in rental costs, some cultural and creative firms chose to withdraw from the old street under the great pressure of maintaining business operations. If the situation continues to deteriorate, the old street is likely to decline in the future.

*5.5. Creation and Operation of Cultural Service Ecosystems in Historical Blocks*

In this study, the operation of cultural heritage service ecosystems was decomposed into actors' activities, the market structure survey, service exchange, cultural value creation, cultural services, brand communication, resource integration, and value co-creation. In terms of the significance level of resource integration and value co-creation, the above elements were categorised into four quadrants. According to the service-dominant logic, services are the basis of exchange and function as entities that apply their knowledge and expertise to benefit another entity [6]. The interaction between two entities leads to the co-creation of services. More specifically, through active participation of actors and the provision of operational resources for exchange, actors continuously exchange services, thus creating value [24]. However, this study found that considering the aspect of actors' activities, the diverse forms of interactions between actors are not necessarily sufficient for meaningful participation, do not necessarily involve a high degree of resource integration, and do not necessarily lead to a consensus. Thus, the importance of their resource integration and value co-creation depends on the nature and degree of the interactions. The market structure survey mostly falls under the individual activities of cultural and creative personnel and is slightly related to other actors. Accordingly, it is an activity with a low significance level of resource integration and value co-creation. In contrast, service exchange between actors necessarily involves an exchange of resources but does not necessarily lead to a consensus. Thus, it is an activity with a high significance level of resource integration. Cultural services and brand communication require close cooperation between actors for service exchange, as well as a high degree of consensus. Thus, they are activities with a high significance level of resource integration and value co-creation. In summary, Figure 2 shows the correlation between the elements of a cultural service ecosystem, resource integration, and value co-creation.

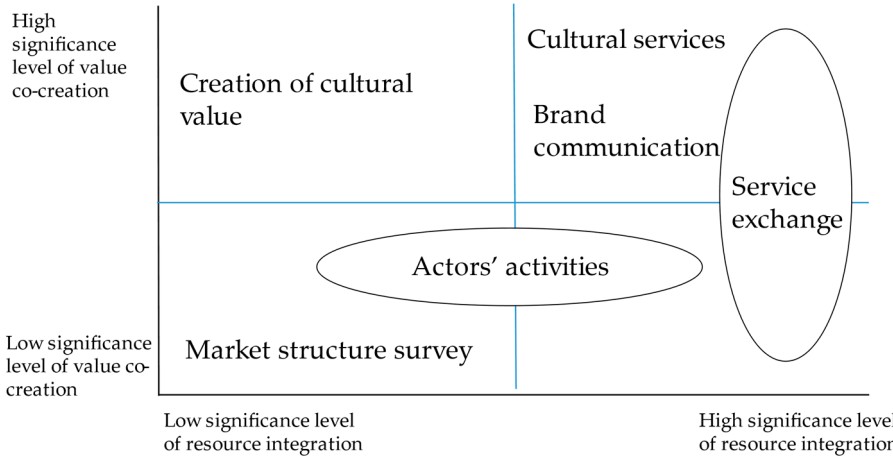

**Figure 2.** Relationship between resource integration and value co-creation in a cultural service ecosystem.

The two cases of this study pertain to the revitalisation of cultural heritage, during which block operation companies were invited, and cultural and creative firms settled in the historical blocks in hopes of reviving the local economy. To achieve sustainable operation and development, develop industrial clusters, and promote the development of local cultural and creative industries, it is imperative that actors in historical blocks adequately exchange services and integrate resources, thus co-creating value for the historic district. Accordingly, this study presents the following proposition:

**Proposition 3.** *Through the survey of market structures and the exchange of services, actors in a historical block fully integrate their resources, co-create cultural value, promote the communication of cultural brands, and provide consumers with sustainable cultural services, thus creating a cultural service ecosystem for the historical block.*

*5.6. Academic Contribution*

In previous studies of cultural heritage, cultural heritage was distinguished into tangible and intangible cultural heritage. Although the concept of 'cultural heritage as a service' has been established internationally, the content and mode of cultural services are severely misconstrued due to the opposition of different standpoints [2]. Such studies presented an 'intangible' perspective for cultural heritage, but they did not investigate how cultural heritage generates cultural value. To generate cultural value, actors in historical blocks must extract cultural meaning from cultural heritage and then present common value propositions, thus co-creating value for historical blocks. This study analysed how actors in historic districts interactively extract and increase the cultural values of historic districts and subsequently develop cultural service ecosystems for historical blocks through resource integration and value co-creation. The findings of this study provide an innovative approach for the sustainable development and economic revival of historical blocks. Today, developing the local economy through the revitalisation of cultural heritage has become a mainstream mode of city promotion worldwide. However, such attempts are mostly limited to the provision of cultural services and rarely focus on developing cultural service ecosystems for historical blocks. This study analysed the operation of cultural service ecosystems of cultural heritage. The findings of this study contribute to the examination of the organisational effectiveness of revitalisation and cultural innovation activities in historical blocks and provide a reference for policymakers pursuing sustainable development of the local cultural economy.

According to the service-dominant logic, actors extract tangible and intangible resources from cultural heritage through active participation and service exchange to increase the cultural value of innovation, whereas cultural service ecosystems are created for cultural heritage. This process also facilitates the creation and development of cultural services, or more specifically, actors co-create services on a consensus basis for their mutual benefit. To ensure that historical blocks are revitalised to provide sustainable cultural services and promote the communication of cultural brands, actors must be able to integrate and share resources. A cultural service system is self-adaptive through mutually beneficial interactions and resource integration [4], thus promoting resource integration [29,30,45]. The degree of resource integration and value co-creation determines whether the revitalisation of historical blocks leads to sustainable operation and whether cultural heritage becomes a cultural brand through the cultural services provided by actors, thus promoting local economic development.

*5.7. Industrial Implications*

Because of the settlement of cultural and creative firms in Dihua Street and great support from public authorities, Dihua Street has been transformed into an internationally known cultural and creative block. This has also resulted in tourism-oriented transformation against the will of local residents. The settlement of Sedai Zone Co. lowered the access threshold for young cultural and creative personnel in Dihua Street. Because of its monopoly of good shops in Dihua Street and the clustering effect, many tourist-oriented cultural and creative firms successively settled in Dihua Street, occupying most of the shops. However, with the short supply of street houses, house rental costs continued to rise sharply, and many shops of traditional Chinese medicine and native products withdrew from Dihua Street. Only self-owned shops survived the rapidly increasing rentals. With a decline in traditional local industries and a reduction in residents' quality of life, the

access threshold for cultural and creative firms continues to increase. Hence, it is a matter of urgency to address these problems and reach a consensus on future steps.

Compared with the development and operation of Dihua Street, Guansi Shihdianzih Old Street encountered various problems (e.g., insufficient publicity by the Hsinchu County government, insufficient public facilities, and poor maintenance and management of old street houses), resulting in small tourist flow and limited revenue from shops. In addition, cultural and creative firms needed to independently repair and maintain old street houses. Moreover, due to heavy economic pressure, most of the cultural and creative personnel had to assume part-time jobs to afford to continue operating their main businesses. The experiential activities held by cultural and creative firms incorporate integrated rich local cultural characteristics. However, it is recommended that local associations have a more comprehensive dialogue with cultural and creative firms, local practitioners, and local residents, integrate their different opinions, give suggestions to public authorities, apply for resource support, and hold various activities suited to the characteristics of various shops to increase the fame of the old street and attract more tourists.

Based on the findings of this study, we offer the following suggestions:

1. Policy aspect: According to Lash and Urry [49], the economy is increasingly influenced by culture that is deeply integrated within the economy. Hence, the boundary between economy and culture has become fuzzy. Therefore, in the process of urban development, it is paramount to develop a culture–economy system integrated within the environment. Taiwanese policy makers have believed that CCIs should be able to create the best economic profit by using a cluster approach. The policy discourses such as creative city, creative economy, and creative class [9,50–52] imply that a cultural heritage cluster would contribute to attracting talent, reimaging, and branding to a city. The central government should encourage the development of ecosystem services on cultural heritage to avoid sectoral fragmentation. Therefore, the public sectors should fully discuss and focus on comprehensive thinking and the integration of resources from cross-sector departments when formulating policies. In addition, cultural and creative firms provide better intermediation between the CCI products as well as their potential markets. They also help compensate for the public sector's weaknesses in terms of business operation and management. In addition to large-scale public construction, county- and city-level governments should allocate more resources to the revitalisation of cultural heritage to, for example, develop the tourism industry, promote the development of the local economy, and implement a balanced approach to urban and rural development.

2. Sociocultural aspect: Cultural services provided by historical blocks are based on the sharing of resources and co-creation of value. Thus, full integration of resources and co-creation of cultural value by actors are paramount to the sustainable operation of cultural brands. In addition, in order to take full advantage of cultural heritage, it is necessary to invest in education and training to the younger generations. This is to develop in them both cultural heritage awareness and skills as well as the competences required in the sector for them. According to the Italian educational system under Law No. 53 and Legislative Decree No. 77, the decree (article 3) stresses 5 significant points [53]:

   (1) Implementing flexible and equivalent learning methods from cultural and educational points of view that compare the outcomes of the second cycle courses. It systematically links classroom training with practical experience.

   (2) Enriching the training acquired in school and training programs with the acquisition of skills that are useful in the labor market.

   (3) Encouraging the orientation of young people to enhance their personal vocations, interests, and individual learning styles.

   (4) Establishing an organic connection of educational and training institutions with the world of work and civil society, which allows the active participation of the subjects involved.

(5)     Correlating the training offered to the cultural, social, and economic development in the territory [53].

It is recommended that the public departments and school institutions under the Taiwan government formulate relevant implementation in which there are measures with reference to the implementation methods of the Italian cultural heritage education system. It would be beneficial to enable students to identify, value, and contribute towards their cultural heritage's dissemination and preservation. In addition, an impact on the community, which conduces creating individual and co-building identities. Thus, contributing towards helping preserve the local cultural heritage can be achieved.

3.     Cultural aspect: Culture has become a major impetus to global urban competitiveness, and the cultural industry has become a critical strategy to increase the benefits of tourism. Culture shapes local economic consumption activities, and accordingly, economic markets contain unique cultural connotations. Historic buildings have great potential for sustainable development based on their unique cultural and economic values. Cultural brands are applicable to cities worldwide, and successful cultural brand cases can be a good example for development planners in other cities.

*5.8. Suggestions for Future Studies and Study Limitations*

Although the cultural industry has become an essential element of the strategy for urban marketing and tourism development worldwide, existing studies on cultural heritage are limited to the restoration of cultural heritage. In the future, it is necessary to intensively investigate the tourism industry facilitated by cultural heritage as well as cultural services provided by cultural heritage. More specifically, it is necessary to explore in-depth how actors can develop complete cultural service ecosystems by creating cultural value, sharing resources, and co-creating value in the case that cultural heritage is being used to develop cultural and creative industries. Globally, cultural heritage and historical buildings can potentially be revitalised and developed in diverse forms. Hence, it is necessary to increase the sample size in subsequent studies and investigate cases of cultural heritage revitalisation performed in different cities and countries and in different forms. The objective is to prevent the inapplicability of findings to other cultural heritage or historical blocks due to the narrow selection of cases. In addition to focusing on the use of cultural heritage and the development of cultural and creative industries, it is worth considering the problems of cultural heritage sites in emergency situations (such as wars, pandemics, or natural disasters) and the difficulties encountered in future research.

**6. Conclusions and Suggestions**

The key factor in successfully revitalising and regenerating cultural heritage lies in whether actors in historical blocks are able to integrate local cultural resources across different fields, promote local economic development, and develop perfect local industry ecosystems. By improving the industry ecosystems, actors can achieve effective complementarity and exchange of resources and develop new business models, thus allowing all participants to benefit. Therefore, actors in historical blocks should co-create symbiotic and win-win cultural service ecosystems. By exploring Dihua Street and Guansi Shihdianzih Old Street, this study investigated the cultural and creative activities of historical blocks and examined how actors should integrate resources to co-create value and cultural service ecosystems should be created and developed in historical blocks.

During the revitalisation of historical blocks and the revival of the local economy, actors integrate resources and co-create value, thus revealing the operation of cultural services for historical blocks. An exploratory study was conducted on Dihua Street and Guansi Shihdianzih Old Street, the results of which provide a theoretical basis for developing cultural service ecosystems for historical blocks. The findings of this study are summarised as follows:

1. Through active participation, exchange of operational resources, and co-creation of services, actors in historical blocks continuously exchange services, thus presenting value propositions.
2. During the revitalisation of cultural heritage and the economic revival of historical blocks, actors exchange services and benefit from interactions by establishing value propositions and surveying market structures.
3. Cultural heritage is revitalised while actors provide cultural services, create cultural value, implement brand communication, and promote local economic development.
4. Through the survey of market structures and exchange of services, actors in historical blocks can fully integrate their resources, co-create cultural value, promote the communication of cultural brands, and provide sustainable cultural services for consumers, thus developing cultural service ecosystems for historical blocks.
5. This study emphasises the importance of sustainable operation of historical blocks as well as resource integration and value co-creation by actors. To achieve sustainable operation of cultural heritage (e.g., historical blocks), it is imperative that actors exchange services effectively, share their resources, and co-create value on a consensus basis.

**Author Contributions:** Writing—original draft, S.-L.K.; writing—review & editing, C.-L.C. All authors have read and agreed to the published version of the manuscript.

**Funding:** This research received no external funding.

**Data Availability Statement:** Not applicable.

**Conflicts of Interest:** The authors declare no conflict of interest.

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
