# Peer review of "Operation Analysis of Cultural Heritage Service Ecosystems: Empirical Study Based on Dihua Street and Guansi Shihdianzih Old Street in Taiwan"

_asi, doi:10.3390/asi5020042_

Round 1
Reviewer 1 Report
Operation Analysis of Cultural Heritage Service Ecosystems: Empirical Study Based on Dihua Street and Guansi Shihdianzih Old Street
by Shih-Ling Kuo , Chun-Liang Chen
Manuscript overview
The manuscript deals with a relevant topic, the development of cultural heritage service ecosystems. To address the topic, the authors discuss an empirical study conducted in two sites in Taiwan.
I find the manuscript quite interesting, but it requires some revisions to increase both the clarity and scientific value of the research as well as its practical implications.
Specific points
1. Introduction
The manuscript lacks both formal definitions of some key concepts and an in-depth state-of-the-art of the topic to which it refers. For example, the manuscript core are cultural heritage service ecosystems, but there is no clear definition of the concept. This will be clarified immediately in the "Introduction" by using relevant literature.
Authors also need to perform an accurate bibliographic analysis on Cultural and Creative Industries (CCI). In fact, there is a lack of both a formal definition of CCI and an analysis of the state-of-the-art of CCI of Taiwan. In order to be able to provide practical usefulness to the research results, these aspects are significant. As regards the state-of-the-art of CCI of Taiwan, I can recommend the works of Pei-Ling & Chapain [1] and Lee [2].
The sentence in lines 36-39 requires reference(s).
1.2. Literature Review
The "Literature Review" should be a section separate from the "Introduction".
The statement in lines 98-99 claims a reference.
I recommend that the authors point out that in order to take full advantage of cultural heritage it is necessary to invest in education and training of the younger generations. This is to develop in them both cultural heritage awareness and skills as well competences required in the sector [3,4].
"Materials and Methods" and "Results and Discussion"
From the point of view of the manuscript structure, I recommend to clarify both the methodology and the research results with flow charts or tables.
Policy aspetcs (section 3???), line 690
The authors could highlight that the resources to be allocated by the central government should encourage the development of ecosystem services on cultural heritage, avoiding sectoral fragmentation.
Furthermore, it would be very helpful to place the research findings in the context of Taiwanese policies. For example, how does the research compare with recent CCI policies of Taiwan? To briefly discuss this point, the authors can take advantage of the works suggested before [1,2].
Minor points
The title of the manuscript must include the country to which the two study sites belong (Taiwan).
All the acronyms must be defined the first time they are mentioned in the text (e.g., line 56, URS)
Concluding, I encourage the authors to perform the revision and submit the updated version of the manuscript.
Kind regards
References suggested
[1] Pei-Ling Liao, Caroline Chapain (2018). Challenges in developing the cultural and creative industries (CCIs) of Taiwan. The issue of local context in cluster policy, in Routledge Handbook of Cultural and Creative Industries in Asia, pp. 72 - 89
[2] https://doi.org/10.1111/ropr.12131
[3] https://doi.org/10.3390/heritage2030120
[4] https://doi.org/10.3390/educsci10070176
Author Response
Dear reviewer
Please see the attachment.
Thank you so much.
Regard,
Shin-Ling Kuo

Reviewer 2 Report
Dear author,
This article contains very important topic of Operation Analysis of Cultural Heritage Service Ecosystems.
The Author correctly summarizes the rules, standards, and directives for the regulation of green areas and points well to the shortcomings in this research topic. The asked main questions were answered, I can accept the thought process. The final conclusions are consistent with the evidence and arguments.
Please complete the literature with items from Europe (Italy, Austria, others). There you can find good examples of activating the heritage.
It is worth considering the problems of heritage sites in emergency situations (war or pandemics or natural disasters).
The work of the author shows that she clearly reviewed and examined the available professional literature data, but references need revision. This paper may be accepted after a minor revision.
Yours Sincerely,
Author Response
Dear reviewer
Please see the attachment.
Thank you so much.
Regard,
Shih-Ling Kuo

Reviewer 3 Report
Dear Authors,
I find this manuscript interesting being as it focuses on how analysis of how cultural service ecosystems on two old streets in Taiwan are created and operated, and how actors develop cultural service ecosystems.
Key words: you should avoid using key words which are also in the title: (cultural service ecosystem).
In abstract is a repetition: Through an exploratory study, this study examined how the cultural service ecosystems of (Line 12)- the second word can be replaced by paper or contribution
Introduction:
It is the same text focused on the description of the two case studies between lines 54-71 and respectively lines 303-320 (methodology: data analysis). You can create a distinct section named the description of case studies after introduction or you should present the case studies as a distinct part of methodology. The description of the two case studies is not a data processing – you can use for the explanation for table 2 a reformulated and adapted text.
1.2. Literature Review
1.2.2 Local culture development: the fragment related to "the current phenomenon of cultural development in Taiwan from the perspective of institutions of cultural and spatial governance..."(Lines 137-142) is not proper located in the section of literature background.
Methodology
You should provide several details about the long-term in-depth interviews: duration of application, a concise presentation of the most important questions.
Results section
I did not find any references about the data obtained after applying the long interviews.
Conclusions:
Some subsections included in Conclusions section should be included in the discussion section:
4.2. Academic contribution
4.3. Industrial implications and
4.4. Suggestions for future studies and study limitations
4.2. Academic contribution - several new ideas can be included: the results should be better compared with the results obtained in other studies by reference to different similarities or differences. This comparison of the results would better reflect the relevance of the study; you should make a better description of the contribution of the results to the development of the field (theoretical or methodological) emphasizing the importance of the study.
Between Lines 626-629 in Academic contribution subsection there is a repetition: In previous studies of cultural heritage, cultural heritage was distinguished into tan-gible and intangible cultural heritage. Although such studies presented an ‘intangible’ perspective for cultural heritage, they did not investigate how cultural heritage generates cultural value.
You can reformulate: In previous studies, cultural heritage was distinguished ....
Author Response

(The authors gave the same response as above.)

Round 2
Reviewer 1 Report
The authors carried out a thorough revision. The work is now clearer and provides useful suggestions. I require three final improvements.
1) In order to emphasize the practical benefits of the research, it would be appropriate to add the following sentence in the abstract (from line 22): "In this perspective, the article provides some useful suggestions for stakeholders and policymakers";
2) Reference 54 does not concern what the authors reported in lines 762-773. Therefore, the authors will replace the reference [54] with the following appropriate reference:
Gizzi, F.T.; Biscione, M.; Danese, M.; Maggio, A.; Pecci, A.; Sileo, M.; Potenza, M.R.; Masini, N.; Ruggeri, A.; Sileo, A.; Mercurio, F.; School-Work Alternation Working Group. Students Meet Cultural Heritage: An Experience within the Framework of the Italian School-Work Alternation (SWA)—From Outcomes to Outlooks. Heritage 2019, 2, 1986-2016. https://doi.org/10.3390/heritage2030120
3) Finally, considering that what the authors report in lines 762-773 are the same words written in the article mentioned just above, it is requested to include points from 1 to 5 on page 18 in quotation marks.
Author Response
Dear Reviewer
Please see the attachment.
Thank you for your all assistance.
Best regard,
Shih-Ling Kuo
